# Molecular Biomarkers for Adrenoleukodystrophy: An Unmet Need

**DOI:** 10.3390/cells10123427

**Published:** 2021-12-06

**Authors:** Madison I. J. Honey, Yorrick R. J. Jaspers, Marc Engelen, Stephan Kemp, Irene C. Huffnagel

**Affiliations:** 1Neurochemistry Laboratory, Department of Clinical Chemistry, Amsterdam University Medical Centers, Amsterdam Neuroscience, Vrije Universiteit, 1081 HV Amsterdam, The Netherlands; m.i.j.honey@amsterdamumc.nl; 2Laboratory Genetic Metabolic Diseases, Department of Clinical Chemistry, Amsterdam University Medical Centers, Amsterdam Gastroenterology Endocrinology Metabolism, University of Amsterdam, 1105 AZ Amsterdam, The Netherlands; y.r.jaspers@amsterdamumc.nl; 3Department of Pediatric Neurology, Emma Children’s Hospital, Amsterdam University Medical Centers, Amsterdam Neuroscience, University of Amsterdam, 1105 AZ Amsterdam, The Netherlands; m.engelen@amsterdamumc.nl (M.E.); i.c.huffnagel@amsterdamumc.nl (I.C.H.)

**Keywords:** biomarkers, biobank, cerebral demyelination, myelopathy, newborn screening, peroxisome, clinical trial, adrenoleukodystrophy

## Abstract

X-linked adrenoleukodystrophy (ALD) is an inherited progressive neurometabolic disease caused by mutations in the *ABCD1* gene and the accumulation of very long-chain fatty acids in plasma and tissues. Patients present with heterogeneous clinical manifestations which can include adrenal insufficiency, myelopathy, and/or cerebral demyelination. In the absence of a genotype-phenotype correlation, the clinical outcome of an individual cannot be predicted and currently there are no molecular markers available to quantify disease severity. Therefore, there is an unmet clinical need for sensitive biomarkers to monitor and/or predict disease progression and evaluate therapy efficacy. The increasing amount of biological sample repositories (‘biobanking’) as well as the introduction of newborn screening creates a unique opportunity for identification and evaluation of new or existing biomarkers. Here we summarize and review the many studies that have been performed to identify and improve knowledge surrounding candidate molecular biomarkers for ALD. We also highlight several shortcomings of ALD biomarker studies, which often include a limited sample size, no collection of longitudinal data, and no validation of findings in an external cohort. Nonetheless, these studies have generated a list of interesting biomarker candidates and this review aspires to direct future biomarker research.

## 1. Introduction

X-linked adrenoleukodystrophy (ALD) is an inherited, progressive, neurometabolic disease which affects the adrenal cortex, testis, and the central and peripheral nervous system. It is the most common leukodystrophy with a birth incidence of 1 in 14,700 live births [1]. ALD results from a mutation to the *ABCD1* gene [2]. This gene encodes the ATP-binding cassette sub-family D member 1 (ABCD1 protein), a peroxisomal transmembrane protein that transports very long-chain fatty acids (VLCFA) as CoA-esters into the peroxisome where they are broken down by peroxisomal β-oxidation [3]. Thus, mutations to the *ABCD1* gene, of which over 900 have been catalogued in the ALD mutation database (https://adrenoleukodystrophy.info/) (accessed on 15 November 2021) [4], result in a deficiency in the ABCD1 protein and impaired degradation of VLCFA [5]. As a result, cytosolic VLCFA levels increase, leading to their storage in plasma and tissues such as the adrenal cortex, the spinal cord, and white brain matter [6,7]. The clinical manifestations of ALD include adrenal insufficiency, myelopathy, and/or leukodystrophy [8]. At birth, individuals with ALD are asymptomatic (Figure 1). Primary adrenal insufficiency is usually the first manifestation of the disease in male patients, with typical onset during childhood [9]. Like adrenal insufficiency, leukodystrophy (often referred to as cerebral ALD) can also manifest during childhood. Cerebral ALD is the most devastating ALD phenotype; if left untreated, the progressive white matter lesions progress and eventually cause severe disability and death [8]. The third and most common clinical manifestation of ALD is progressive myelopathy (also referred to as adrenomyeloneuropathy, AMN), which occurs in almost all male ALD patients [10], and 80% of female patients [11,12].

## 2. Molecular Biomarkers for ALD: An Unmet Need

As defined by the FDA-NIH Biomarkers Definitions Working Group, a biomarker is a “characteristic that is measured as an indicator of normal biological processes, pathogenic processes, or biological responses to an exposure or intervention, including therapeutic interventions” [13]. Biomarkers can be categorized into different types depending on their use; for example, diagnostic biomarkers, which are used to detect the presence of a disease or subtype of the disease, or prognostic biomarkers, which inform about the likelihood of a medical incident, recurrence, or progression of the disease in patients. Other biomarker categories include susceptibility/risk biomarkers, monitoring biomarkers, predictive biomarkers, pharmacodynamic/response biomarkers, and safety biomarkers.

All patients have a mutation in the *ABCD1* gene with mutations ranging from missense mutations to large deletions. Despite being a monogenetic disease, mutations in the *ABCD1* gene have no predictive value with respect to clinical outcome. The lack of a simple genotype-phenotype correlation can be exemplified with a case study detailing six brothers with an identical p.Pro484Arg *ABCD1* missense mutation, yet presenting with five different ALD phenotypes (ranging from cerebral ALD in childhood to adrenal insufficiency in adulthood) in the family [14]. Furthermore, even monozygotic twins have been reported with only one sibling affected by cerebral ALD [15,16]. Clinical manifestations of the disease are most likely determined by a combination of genetic, epigenetic, and environmental factors [8,17]. Therefore, ALD biomarkers are needed to better understand and predict the progression of the disease, as well as to serve as surrogate outcome measures for clinical trials. Additionally, it would be helpful to have measures of disease activity that are usable for asymptomatic as well as severely affected patients (i.e., no floor and ceiling effect) so that more patients are eligible for clinical trials. In particular, new ALD biomarkers are needed:(1)As a diagnostic marker to identify the onset of leukodystrophy in patients with ALD. Currently, an MRI of the brain is used to detect white matter lesions in an asymptomatic stage so that treatment (hematopoietic cell transplant; HCT) is still an option [18,19]. However, an MRI of the brain is relatively costly and for young children invasive because of the need for general anesthesia.(2)As a predictive marker to identify individuals at high risk for cerebral ALD before its onset, particularly with respect to newborn screening positives [20,21]. Currently, all male ALD patients undergo the same intensive follow-up [22,23,24]. However, only an estimated 40% will develop cerebral ALD before the age of 18 years. If markers were identified to stratify patients in “high” and “low” risk groups for the development of cerebral ALD, personalized follow-up could be offered.(3)As a prognostic biomarker to predict fast versus slow progression of myelopathy is needed as this would allow for: (1) better counselling of individual patients and (2) for the stratification of patients in clinical trials.(4)As a disease monitoring biomarker that correlates with disease severity (for example the myelopathy) and is dynamic (i.e., changes within a reasonable timeframe as the disease progresses). This would be extremely useful as a surrogate.

This review focuses on molecular biomarkers as they may have a (causal) role in disease development, thus providing more insight into disease pathology, since current clinical outcome measures have limitations that make performing clinical trials challenging. For example, the Expanded Disability Status Scale (EDSS) is the most commonly used measure to assess disability in studies concerning myelopathy in ALD patients [10], however, it is subject to high variability in inter-rater and intra-rater reliability [25]. Additionally, recent developments in monitoring disease progression in ALD using optical coherence tomography and body sway are promising, but still require trials of several years duration and a large number of participants [26,27]. The increasing amount of biological sample repositories (‘biobanking’) creates a unique opportunity for rapid evaluation of new or existing biomarkers. As biomarkers need validation in an external cohort before they can be implemented, this review aspires to direct future biomarker research by recommending a panel of biomarkers based on a literature review.

## 3. Setting the Stage

In order to identify and validate ALD biomarkers, follow-up and biobanking over many years to decades is necessary in long observational studies which involve patient care and research. The introduction of newborn screening provides an ideal opportunity, since follow-up of newborns can begin in the asymptomatic stage of the disease. To evaluate potential new predictive and disease monitoring biomarkers, these will have to be correlated to existing outcome parameters, which change very slowly over time [10]. Due to the variability of symptoms and slow progression, many patients will have to be followed for a very long time, making these studies expensive and logistically challenging. Biobanks only have value for this kind of research if samples are collected and stored in a standardized way in a prospective manner and can be linked to detailed and accurate clinical information of individual patients. Although for ALD a few of these cohorts exist, these are all local efforts in individual centers. Larger international cohorts and biobanks would be preferable but have not materialized even though researchers recognize the potential benefits. Major obstacles are complex regulatory issues and ownership of samples. An alternative would be to maintain individual cohorts and biobanks, but with a standardized follow-up protocol to make the individual data sets more compatible. It will be interesting to see how the field will work towards better collaboration.

## 4. Biomarker Studies So Far

### 4.1. Very Long-Chain Fatty Acids

Since VLCFA accumulation is the primary biochemical abnormality in ALD, various studies have investigated the possible correlation between VLCFA levels and clinical outcomes. In 1984, Moser et al. reported on the VLCFA analysis in plasma and fibroblasts from 303 ALD patients with a wide phenotypic variation [28]. No correlation was found between C26:0 levels or the C24:0/C22:0 or C26:0/C22:0 ratio and disease severity. Furthermore, no differences in VLCFA levels were observed in asymptomatic patients and their symptomatic relatives. Similar findings were reported by Boles et al. in 1991 [29], who demonstrated no differences in VLCFA levels or the fatty acid composition of isolated sphingomyelin and phosphatidylcholine factions in fibroblasts from four myelopathy and six cerebral ALD patients [29]. The findings of Moser et al. and Boles et al. were confirmed by a 1999 study from Moser et al. [30]. In this study, plasma VLCFA levels of 1097 ALD males were analyzed. The distribution of VLCFA in plasma of boys with cerebral ALD was identical to adult patients with myelopathy. Stradomska et al. reported the VLCFA levels in 127 ALD patients [31]. VLCFA levels did not differ with age or disease severity. In 1991, Antoku et al. reported a significantly higher C26:0/C22:0 ratio in the mononuclear cells of four patients with cerebral ALD in childhood (age: 7–18) compared to four patients with cerebral ALD in adulthood (age: 31–39) [32], whereas there was no difference in the C26:0/C22:0 ratio in erythrocyte membranes and blood plasma between the two groups. The small sample size, however, limits the interpretation of these results.

It is well established that plasma VLCFA levels are not predictive for the development of cerebral ALD. However, various studies have investigated the role of VLCFA and VLCFA incorporated lipids in the brain in the initiation of cerebral ALD. Analysis of postmortem brain samples of cerebral ALD patients demonstrated that in normal appearing white matter, VLCFA accumulates primarily in the phosphatidylcholine fraction [33]. In actively demyelinating regions, VLCFA were strongly elevated in cholesterol esters [33,34]. These cholesterol esters could primarily be located in macrophages and microglia which are involved in the inflammatory demyelinating process after disruption of the blood brain barrier [35]. One of the few studies that demonstrated a possible relationship between VLCFA levels in brain and ALD phenotype is a study by Asheuer et al. [36]. In this study, the authors measured the VLCFA levels in normal appearing white matter from controls, patients with cerebral ALD in childhood, adult myelopathy patients, and myelopathy patients that had subsequently developed cerebral demyelination. Compared to controls, all ALD patients had increased levels of saturated VLCFAs (C26:0–C30:0) in normal appearing white matter. Interestingly, higher saturated VLCFA levels were found in normal appearing white matter from the childhood cerebral ALD group compared to the myelopathy and myelopathy-cerebral group. Moreover, the levels of the mono-unsaturated fatty acids C22:1, C24:1, and C25:1 were decreased in all cerebral ALD patients but not in myelopathy patients compared to controls. These results indicate that saturated VLCFA levels may play a role in the development of cerebral ALD. In fact, there is evidence suggesting VLCFA accumulation is related to inflammatory cytokine expression in the pathogenesis of cerebral ALD [37]. The possible importance of the balance between saturated and mono-unsaturated VLCFA as a contributing factor deserves further study. For example, cell studies demonstrated that exposure of ALD fibroblasts to saturated VLCFA (C26:0) induced endoplasmic reticulum (ER) stress and lipid-induced cell death, whereas exposure to the mono-unsaturated VLCFA (C26:1) did not [38]. Furthermore, metabolic rerouting of fatty acid synthesis by pharmacological induction of stearoyl-CoA desaturase (SCD1) activity shifted the balance towards mono-unsaturated VLCFA, thereby normalizing VLCFA levels both in vitro and in Abcd1-deficient mice [39]. From these studies, it is apparent that peripheral VLCFA levels (i.e., in plasma, blood cells, or fibroblasts) are of diagnostic value for ALD, however they do not reflect VLCFA metabolism or levels in the brain. Therefore, using a lipidomic approach in the CSF may identify lipids or a lipidomic signature which better reflect the changed lipid levels in the brain.

### 4.2. Complex Lipids and Lysophosphatidylcholine (lysoPC)

Due to the lack of prognostic value of VLCFA, various efforts have been made to identify lipid biomarkers that correlate with disease form and/or disease severity. In 1976, Igarashi et al. demonstrated that VLCFA are incorporated in a variety of lipids including gangliosides, phosphatidylcholines, and cholesterol esters [6]. Phosphatidylcholines (PC) can be converted to lysoPC by the enzymatic action of phospholipase A2 (PLA2) and/or lecithin-cholesterol acyltransferase (LCAT) [40]. The VLCFA containing C26:0-lysoPC is elevated in all ALD men and women and is used in ALD newborn screening [1,22,41,42]. Recently, it was demonstrated that women with ALD with plasma VLCFA levels in the normal range have elevated levels of C26:0-lysoPC in dried blood spots and plasma [43,44]. Thus, C26:0-lysoPC outperforms VLCFA analysis as an ALD diagnostic biomarker. Currently there is no evidence that C26:0-lysoPC levels in plasma or dried blood spots are predictive concerning disease progression or manifestations. Whether C26:0-lysoPC levels in the brain correlate with phenotype, as has been demonstrated for VLCFA levels [36], has not yet been investigated. Although it is currently unclear how C26:0-lysoPC levels relate to ALD pathogenesis, there is evidence that lysophosphatidylcholines could play a role in the development of neurodegenerative diseases [40]. In fact, injection of C24:0-lysoPC, but not C16:0-lysoPC, into the parietal cortex of wildtype mice resulted in widespread microglial activation and apoptosis indicating a potential role for VLCFA-containing lysoPCs in the pathogenesis of cerebral ALD [45]. Interestingly, a recent study by Kettwig et al. reported lower levels of C20:3-lysoPC and C20:4-lysoPC in serum from cerebral ALD patients compared to neurologically asymptomatic ALD patients [46]. Moreover, the levels of these lipids were even lower in the serum of cerebral ALD patients approximately one year prior to the first changes in their cerebral MRI. If these findings are confirmed in an external cohort, C20:3-lysoPC and C20:4-lysoPC levels may be indicative of the neuroinflammation in ALD patients.

One approach that is increasingly used for the identification of new biomarkers is lipidomics [47]. The aim of lipidomics is the unbiased analysis and study of the complete lipid content in a biological system using analytical chemical techniques. In 2019, Huffnagel demonstrated that a semi-targeted lipidomics analysis could be used to identify new candidate lipid biomarkers in women with ALD [12]. Thus far, three studies have used lipidomics in an attempt to identify phenotype-specific markers. Lee et al. performed a lipidomics analysis on fibroblasts from four cerebral ALD patients, three myelopathy patients, and two healthy controls [48]. The lipidomics analysis revealed lower triacylglycerol and ceramide levels in the cerebral ALD group compared to the myelopathy group. However, the limited number of samples and the inclusion of up to seven biological replicates for each cell line hinders the interpretation of these findings. Fujiwara et al. performed a lipidomics analysis with a focus on glycosphingolipids on fibroblasts from four childhood cerebral ALD patients, three myelopathy patients, and five healthy controls [49]. Several VLCFA-containing glycosphingolipid and sphingomyelin species were elevated in fibroblasts from ALD patients compared to controls. Interestingly, only hexosylceramide 44:1 (HexCer 44:1) showed a significant difference between the childhood cerebral ALD and myelopathy fibroblasts. Considering the important role of glycosphingolipids in the nervous system, it will be interesting to see whether this finding can be replicated in a larger cohort. Richmond et al. employed a lipidomics strategy using plasma samples of six well-characterized brother pairs affected by ALD and discordant for the presence of cerebral ALD [50]. The study was unable to separate cerebral ALD and non-cerebral ALD using principal component analysis which indicated very similar lipid profiles of these two groups and no statistically significant lipid marker was identified that could discriminate between cerebral ALD and non-cerebral ALD patients. Likely reasons for this negative result may have been the small cohort with only six discordant sibling pairs as well as the possibility that during follow-up some patients will have to be reclassified as cerebral ALD.

### 4.3. Oxidative Damage Markers

When determining which molecules to explore as ALD biomarkers, understanding their biochemical function and therefore, involvement in the disease, provides rationale for their investigation and clinical translation. Oxidative damage has been shown to occur before disease onset in a mouse model of ALD [51], and is believed to underlie axonal degeneration, since treatment with antioxidants could counteract this damage [52] as well as reverse locomotor impairment in mice deficient in both *ABCD1* and *ABCD2* [53]. Since glutathione plays an essential role as an antioxidant in the protection of cells against oxidative stress [54], Petrillo et al. evaluated the lymphocyte, erythrocyte, and plasma concentrations of oxidized and reduced glutathione in 14 ALD patients (four with cerebral ALD in childhood and ten adults with myelopathy) compared to age-matched controls, using HPLC and spectrophotometric assays [55]. They discovered that the levels of reduced glutathione (GSH) in erythrocytes, plasma, and lymphocytes were significantly decreased in myelopathy patients compared to controls. Cerebral ALD patients were found to have a significantly lower concentration of GSH compared to controls in erythrocytes only, however this could be due to the extremely small number of cerebral ALD patients in the study and thus a lack of statistical power [55]. A study from Nury et al. employed a commercially available colorimetric assay to measure GSH in the plasma of 17 ALD patients (two boys with cerebral ALD, two adults with cerebral ALD, three patients with adrenal insufficiency, and ten adult myelopathy patients) compared to ten healthy controls and confirmed the findings from the Petrillo study [56]. In addition, the study found decreased levels of docosahexaenoic acid (C22:6), an omega-3 poly-unsaturated fatty acid required for neuronal function and signaling [57], and α-tocopherol, an active form of vitamin E which functions to protect against oxidative damage [58] in ALD patients compared to controls [56]. These studies support the concept that oxidative stress is a contributing pathogenic factor in ALD. Despite these two independent studies supporting the same finding that reduced GSH is decreased in ALD patients compared to controls, the lack of longitudinal data, correlation of GSH measurement to any clinical parameters of disease severity, and validation in external cohorts prevents a definitive verdict on glutathione being used as a biomarker for ALD. It would also be interesting to include asymptomatic ALD patients in these studies and monitor any changes in GSH levels over time, particularly when symptoms develop.

In relation to the proposed redox imbalance involved in the pathophysiology of ALD [59], Casasnovas et al. performed a phase II pilot, open-label study (ClinicalTrials.gov Identifier: NCT01495260) in which 13 subjects (12 men and one woman) with myelopathy were given a high dose combination of three antioxidants (α-tocopherol, N-acetylcysteine, and α-lipoic acid). The primary outcome was the validation of a set of biomarkers for monitoring the biological effects of this antioxidant cocktail [60]. Patients with cerebral ALD were excluded from the study. The age of the 13 myelopathy patients ranged from 24 to 64 years, with 25 healthy, age- and sex-matched controls used for plasma, urine, and peripheral blood mononuclear cell (PBMC) measurements. The study also included functional clinical scales, the 6 min walk test (6MWT), electrophysiological studies, and cerebral MRI as secondary outcome measures to assess any clinical improvement from the antioxidant treatment and any correlations with biomarker measurements. After three months of antioxidant treatment, the levels of pro-inflammatory markers (IFNA2, IL-4, IL-36A, and CCR3 in peripheral blood mononuclear cells, and 12S-HETE, 15S-HETE, TXB2, TNF, and IL-8 in plasma) decreased significantly and the plasma level of the anti-inflammatory marker adiponectin increased. Furthermore, the levels of all oxidative damage markers investigated (N^ε^-carboxymethyl-lysine (CML), N^ε^-carboxyethyl-lysine (CEL), N^ε^-malondialdehyde-lysine (MDAL) and aminoadipic semialdehyde (AASA) in plasma and 8-oxo-dG in urine) significantly decreased three months after treatment. Importantly, none of the biomarkers measured at baseline in the study significantly correlated to the clinical status of patients and therefore would not be suitable as disease-monitoring biomarkers. Although the study by Casasnovas et al. reported no significant difference in nerve conduction studies, EDSS score, or MRI outcomes one year after antioxidant treatment, there was a significant improvement in the distance walked in the 6MWT in eight out of the ten patients who completed the study, as well as an overall significant decrease observed in central motor conduction time (CMCT) in both legs of patients. Two of the inflammatory markers tested were predictive of the response to treatment. First, the ratio of monocyte chemoattractant protein 1 (MCP-1, a pro-inflammatory marker) levels at month seven compared to baseline levels (M7/M0), displayed a significant predictive value for the 6MWT result at the end of the study (21 months). Biomarker values were measured at month seven since 12 out of the 13 patients initially received a low dose of antioxidants for two months, followed by a washout period of two months, then an increased dosage of antioxidants for three months (totaling 7 months). Second, the levels of 15S-HETE (a pro-inflammatory marker) at month seven also displayed predictive value for the 6MWT at the end of the study (Pearson correlation coefficients of 0.97 for the MCP-1 ratio and 0.95 for 15S-HETE). Considering this study, myelopathy in ALD potentially has an inflammatory component. However, additional cohorts with greater patient numbers are required to confirm the predictive value of MCP-1 and 15S-HETE. The strengths of this study were that the pre-treatment concentrations of oxidative stress and inflammatory markers in patients were compared to controls, as well as longitudinal data being collected to assess if levels change within patients during and after treatment. However, the study did not include an untreated control group and thus, there is a possibility of a placebo effect. Therefore, this panel of biomarkers should be examined in future clinical trials investigating the use of antioxidants with larger patient numbers, a longer study period and placebo controls.

A paper from Turk et al. investigated the plasma activity of superoxide dismutase (SOD) in a cohort of ALD patients with different phenotypes; ten males with myelopathy, eight males with cerebral ALD, three female ALD patients, and nine age-matched healthy controls [61]. SOD is an enzymatic antioxidant which provides cellular protection against reactive oxygen species, and polymorphisms of SOD have been associated with the cerebral ALD phenotype [62], therefore this was the rationale for its investigation as a biomarker. Most importantly, the study had longitudinal data analyzed from four myelopathy patients who subsequently developed cerebral ALD during the study, with multiple simultaneous MRI and blood samples obtained preceding cerebral ALD onset [61]. In all four patients, a reduction in SOD activity over time occurred, suggesting that a decrease in SOD activity occurs prior to the onset of cerebral ALD. However, more longitudinal data is required to reveal if the rate at which SOD activity decreases varies with symptom progression or if SOD activity could be a prognostic or diagnostic biomarker for cerebral ALD, in addition to data on the natural variability in patients. Although the study included a range of different ALD phenotypes, it is unfortunate that the authors considered females as carriers of the disease only, and thus did not state any symptoms or possible ALD phenotype the females may have had due to their ALD diagnosis. Nevertheless, they found that increasingly severe phenotypes showed a significant decrease in plasma SOD activity, where the controls had the highest SOD activity, followed by female ALD patients, males with myelopathy, then cerebral ALD patients. The strength of this study is that this result was validated in a second cohort using stored biobank samples of ten male myelopathy patients and 20 cerebral ALD patients, showing a significant reduction between the myelopathy and cerebral ALD SOD levels in both fresh and stored samples. Additionally, the authors investigated if measurements of SOD activity were influenced by long-term storage and freeze-thaw cycles of samples and found that the intraindividual biological variability of plasma SOD was a maximum of 15% over six months, suggesting it is suitable for use as a biomarker [61]. This result contradicts the finding of Petrillo et al., who found no significant difference in SOD activity between myelopathy, cerebral ALD, or control samples [55], however this could be due to the difference in assays used in both studies to measure SOD activity, and that Petrillo et al. assayed activity in hemolyzed erythrocytes rather than plasma [55]. Furthermore, the data of Turk et al. is in line with the clinical MRI severity score, which shows that the most severely affected patients have lower SOD activity in both the biobank and fresh plasma samples of cerebral ALD patients [61].

### 4.4. Inflammatory Markers

Various studies have demonstrated the role of inflammation in ALD disease pathology. VLCFA accumulation provokes a neuroinflammatory response in which immune cells such as macrophages and T cells secrete cytokine signaling proteins, which have been linked to inflammatory demyelination [63]. Marchetti et al. therefore aimed to investigate a selection of pro- and anti-inflammatory mediators in the plasma of ALD patients with different ALD phenotypes [64]. The study included only male patients; eight with childhood cerebral ALD, eight adults with myelopathy, and eight asymptomatic patients. However, only eight healthy controls were used and therefore samples were not age-matched. The study reported significantly higher levels of both pro-inflammatory (IL-1β, IL-8, TNF-α, and IL-2) and anti-inflammatory (IL-10 and IL-4) cytokines in asymptomatic patients compared to controls which were stated to have no correlation with age. Myelopathy patients had elevated levels of the pro-inflammatory markers TNF-α and IL-2 and anti-inflammatory markers IL-4 and IL-5 compared to controls. No significant differences in biomarker levels were found between the childhood cerebral ALD cases and controls [64]. The study lacked follow-up data on any of the patients and is therefore not useful in assessing the potential of any inflammatory cytokines as predictive or treatment-responsive biomarkers for ALD. The authors speculated that the higher levels of these inflammatory markers could be an early marker of disease, particularly since the anti-inflammatory marker IL-10 was significantly elevated in the asymptomatic group compared to all other groups and therefore may represent an early-diagnostic biomarker before the onset of clinical symptoms. The pro-inflammatory chemokine IL-8 was also investigated in a study from Cappa et al., however it was undetectable in the CSF of ALD patients [65]. This was surprising given that Marchetti et al. could detect IL-8 in plasma [64] and since CSF surrounds the brain and spinal cord directly, biomarker concentrations are expected to mirror the underlying inflammation occurring in the brain [66]. This disparity could be due to the sensitivity of the detection methods used in the studies, or that IL-8 levels in blood represent a systemic marker of inflammation as opposed to neuroinflammation specifically. Both CSF and blood have advantages and disadvantages as biological fluids for biomarker discovery; CSF is collected by a lumbar puncture procedure which is highly invasive when compared to blood collection, and this can limit its use in a clinical environment. Additionally, the study of Cappa et al. examined the CSF levels of the pro-inflammatory cytokine IL-6 in five female ALD patients before and after administration of a mixture of Lorenzo’s oil and conjugated linoleic acid for a two-month time period [65]. The researchers chose IL-6 since it is a key mediator of the acute phase response which occurs soon after the onset of inflammation [67]. Using a sandwich immunoassay the researchers found that a decrease in IL-6 was only observed in three out of five patients at the end of the study [65]. Although the study recruited female ALD patients whom are largely overlooked in most papers, the very small cohort size and lack of correlation with any clinical outcome measures prevented this study from being of use in assessing IL-6 and IL-8 as viable biomarkers. A more extensive study conducted by Lund et al. evaluated 23 inflammatory factors, including IL-8, in CSF and serum samples of 36 boys with cerebral ALD, prior to them receiving HSCT [68]. This study was able to detect IL-8 in CSF, and indeed found that levels of IL-8 in addition to IL-1ra, monocyte chemoattractant protein-1 (MCP-1), macrophage inflammatory protein 1beta (MIP-1b), and vascular endothelial growth factor (VEGF) in cerebral ALD patients were all significantly higher than in controls. However, in serum, only stromal cell-derived factor 1 (SDF-1) concentration was significantly elevated in cerebral ALD compared to control cases. The control group used consisted of boys at least three months into maintenance therapy for acute lymphoblastic leukemia, with no CSF leukemia present. They were chosen since lumbar punctures are routine every three months during their maintenance therapy, and the risk of carrying out this procedure on “healthy” children was deemed too great and ethically challenging. Although this can be considered a limitation of the study, the researchers note that CSF cytokine levels in ‘control’ children undergoing routine diagnostic lumbar puncture for undisclosed reasons reported in previous literature were highly similar to their data [69]. The strengths of this study included the large cohort size of cerebral ALD patients, the measurement of each sample in duplicate, and the use of previously validated commercially available detection procedures to obtain average cytokine concentrations. Furthermore, the study correlated cytokine measurements to a pre-HSCT MRI severity score. The MRI severity score is an MRI-based quantification of the amount of cerebral involvement [70]. Regression analysis showed that levels of serum SDF-1 (R2 = 0.33, *p* = 0.003), and CSF IL-8 (R2 = 0.12, *p* = 0.04) and MCP-1 (R2 = 0.19, *p* = 0.008) all significantly correlated with the pre-HSCT MRI severity score of patients [68]. The study also demonstrated that total protein levels in CSF were significantly increased in cerebral ALD patients compared to controls, and significantly correlated with CSF IL-8 and MCP-1b levels, as well as the MRI severity score [68], a finding which they had previously reported in a separate cohort of 25 boys with cerebral ALD [69]. These results suggest that MCP-1 and IL-8 levels in CSF may be useful as a predictive biomarker of clinical progression and/or an early indicator of neuroinflammation in cerebral ALD, and thus warrants further investigation. Interestingly, levels of the CSF cytokine MCP-1 were also found to strongly correlate with chitotriosidase activity [68], which is an enzyme secreted by activated macrophages which is elevated in certain lipid storage lysosomal diseases [71]. Another study from the researchers at the University of Minnesota reported that chitotriosidase activity was significantly increased in both the plasma and CSF of 38 cerebral ALD patients compared to 16 controls [72]. Furthermore, there were significant correlations between CSF and plasma chitotriosidase activity and pre-HSCT neurological functional scores (R2 = 0.1742, *p* = 0.01 for CSF and R2 = 0.4025, *p* < 0.0001 for plasma), one-year post-HSCT neurological functional scores (R2 = 0.6514, *p* < 0.0001 for CSF and R2 = 0.4666, *p* < 0.0001 for plasma), as well as CSF chitotriosidase activity significantly correlating with one-year post-transplant MRI severity scores (R2 = 0.3492, *p* = 0.0004). Chitotriosidase activity may therefore be of value as a prognostic biomarker for cerebral ALD patients undergoing HSCT, particularly since a recent study from the same group has provided supporting evidence in a larger cohort of cerebral ALD patients [73]. Plasma and CSF chitotriosidase activity was analyzed in a cohort of 66 boys with cerebral ALD who underwent successful HSCT and found that both pre-transplant plasma and CSF chitotriosidase activity were significantly associated with the volume of gadolinium enhancement on MRI 30 days post-HSCT (univariate *p* value = 0.04 for both plasma and CSF). Since this early gadolinium resolution following HSCT is associated with a reduced neurologic progression one-year post-HSCT, chitotriosidase holds promise as a highly sensitive prognostic biomarker for cerebral ALD patients undergoing HSCT.

The neuroinflammatory process thought to drive cerebral ALD is associated with blood-brain-barrier (BBB) disruption. Lund et al. hypothesized that BBB disruption may occur through the degradation of the extracellular matrix defining the BBB capillary network by matrix metalloproteinases (MMPs) [68,69]. The concentration of MMPs and tissue inhibitors of metalloproteinases (TIMPs), which directly inhibit MMPs, were determined in the CSF and plasma of 20 boys with cerebral ALD prior to HSCT and compared to 19 control patients who were undergoing intrathecal chemotherapy as treatment for a prior diagnosis of acute lymphoblastic leukemia and were without CSF leukemia [68,72]. In addition to correlation of total protein levels in the CSF to MRI severity scores, the study also evaluated if any MMP or TIMP levels correlated to pre- and post-HSCT neurologic function scores. CSF levels of both MMP10 and TIMP1 significantly correlated with the pre-transplant MRI severity scores. TIMP1 and MMP10 correlated significantly with the pre-transplant neurologic functional scores and one-year post-transplant neurologic functional score, respectively. CSF total protein levels were significantly increased in the cerebral ALD patient group compared to controls (*p* < 0.0001) [69]. Furthermore, when compared with MMPs and TIMPs, total protein concentration in the CSF showed superior correlation to the pre-transplant MRI severity score (*p* = 0.0003, R2 = 0.55), and thus the severity of neuroinflammation, in addition to the neurologic functional score pre-treatment (*p* < 0.001, R2 = 0.48) and one year post-HSCT (*p* < 0.0001, R2 = 0.69) [69]. This neurologic functional score assesses dysfunction in several different areas such as vision, hearing, speech and gait, thus these results suggest that CSF total protein levels may be the best marker of gross disease progression and severity, as well as post-HSCT outcome for cerebral ALD patients. However, future research should investigate if CSF total protein displays a prognostic ability for different outcome measures following HSCT in cerebral ALD patients. In relation to the proposed function of MMPs in cerebral ALD, results from Orchard et al. suggest that APOE4 may act as a disease modifier during the course of cerebral ALD, enhancing inflammation and blood brain barrier disruption via a biochemical pathway involving MMP2, resulting in a more severe cerebral disease. Proteomic analysis of pooled CSF from 18 young boys with active cerebral ALD prior to HSCT was performed in the study [74]. Comparison with pooled non-ALD CSF from 19 young boys prior to intrathecal chemotherapy as treatment for acute lymphoblastic leukemia [68,72] identified APOE4 as a possible risk factor. Apolipoprotein E (ApoE) is a major lipoprotein that functions in lipoprotein-mediated lipid transport between organs, both in the periphery and in the central nervous system. Different genotypes in combination with two SNP sites in the ApoE gene produce three isoforms, APOE2, APOE3, and APOE4. Next, the CSF of 21 young males with cerebral ALD who carried the APOE4 allele was compared to 62 non-carriers with cerebral ALD. Frequency of APOE4 inheritance was slightly higher (17.5%) in cerebral ALD patients than in the general population. Young males with cerebral ALD carrying APOE4 had a higher gadolinium intensity score (2.0 vs. 1.3 points, *p* = 0.007), more neurologic impairment (neurologic function score 2.4 vs. 1.0, *p* = 0.001), and a 50% higher MRI severity score, which indicates a higher cerebral disease burden at the time of evaluation for HSCT [74]. Validation in an independent cohort and a longitudinal study including ALD patients before the onset of cerebral ALD is required to better understand the role of APOE4 during pathology manifestation and to determine if it truly has disease-modifying action.

Importantly, all studies from the University of Minnesota group included cerebral ALD patients who already had abnormal MRIs at the time of biofluid collection. If these findings can be validated, it would be beneficial to sample patients prior to the onset of cerebral ALD to investigate if an increase in these cytokines, MMPs, and chitotriosidase, particularly MCP-1, occurs before, or as a consequence of, the neuroinflammation and damage to the BBB, and if it could be an early predictive biomarker for cerebral ALD.

Demyelination, neuroinflammation, and BBB disruption are shared characteristics with a variety of auto-immune diseases. In some, a specific antibody may be a mediator of disease, in others, an antibody can serve as biomarker of disease. To investigate whether the onset of cerebral ALD is also associated with specific auto-antibodies, plasma samples from boys with ALD but without cerebral ALD (n = 29), boys with cerebral ALD (n = 94), and control boys without ALD (n = 30), were analyzed [75]. This analysis showed that autoantibodies against profilin 1 (PFN1) were overrepresented in boys with cerebral ALD. Anti-PFN1 antibodies were present in 0/30 controls, 2/29 (7%) boys with ALD but without cerebral ALD, and 48/94 (51%) boys with cerebral ALD. In a separate study, the CSF of 22 control boys and 15 boys with a new diagnosis of cerebral ALD was analyzed. This showed that both the concentration of PFN1 and plasma cytokine B-cell activating factor (BAFF) were significantly increased in cerebral ALD patients compared to controls. The presence of anti-PFN1 antibodies correlated with the gadolinium intensity score [76], but did not correlate with clinical scores and described biomarkers of cerebral ALD including: age of assessment, MRI severity score, neurologic functional score, CSF total protein, plasma or CSF chitotriosidase activity, or CSF MMP2 activity [75]. This was unexpected since the gadolinium intensity score has been shown to positively correlate with CSF chitotriosidase activity [76]. The lack of correlation may be due to the use of two different detection methods to measure anti-PFN1 antibody levels [75]. Future work should use a quantitative and more sensitive assay to determine the levels of anti-PFN1 antibody in boys with ALD but without cerebral ALD, and those who have undergone HSCT. Assay development and validation would be a large hurdle to overcome but is essential to establish if anti-PFN1 antibodies are of use as an early biomarker of cerebral ALD and/or an outcome measure for cerebral ALD treatments. Using a multi-omics approach in six well-characterized brother pairs affected by ALD and discordant for the presence of cerebral ALD, Richmond et al. confirmed the PFN1 protein overabundance in four out of six cerebral ALD patients [50]. Since MRI severity scores were not reported, conclusions cannot be drawn as to why cerebral ALD patients do not all have a ubiquitous PFN1 phenotype. Future work could investigate in a larger cohort the levels of plasma PFN1 including in post-HSCT patients, as well as before the onset of cerebral ALD, to observe if an increase in PFN1 protein levels precedes the autoimmune response within the cerebral ALD patients who demonstrate PFN1 autoreactivity.

### 4.5. Neuro-Axonal and Astroglial Injury Markers

The importance of utilizing longitudinal data was shown in publication by van Ballegoij et al. [77]. The authors investigated the use of plasma neurofilament light (NfL) and glial fibrillary acidic protein (GFAP) as biomarkers for spinal cord degeneration in a cohort of 45 male and 47 female ALD patients [77]. Compared with a cohort of 74 healthy controls, plasma NfL and GFAP levels were elevated in both symptomatic and asymptomatic male and female ALD patients. The study also investigated correlations between plasma NfL and GFAP levels and three clinical measures of myelopathy severity; the EDSS, the severity scoring system for progressive myelopathy (SSPROM), and the timed up-and go test. Importantly, the paper adjusted for the confounding effect of age on NfL and GFAP measurements, using multiple regression analysis. GFAP did not show any correlation with clinical severity. In contrast, NfL correlated with all three clinical parameters in male patients where the more severely affected patients had increased plasma NfL levels, however this trend was not observed in female patients. Using longitudinal data from 18 male patients in the study, collected over a timespan of two years, the researchers found no correlations between changes in NfL concentration and clinical parameters measuring disease progression [77]. This could be due to the slowly progressive nature of the disease, coupled with the small cohort number and low sensitivity of the clinical parameters in measuring myelopathy progression. The researchers hope to overcome this issue through the continued collection of longitudinal data from the same cohort. In support of these findings regarding NfL, Weinhofer et al. measured blood NfL in 94 ALD patients and 55 healthy volunteers, totaling 199 samples [78]. Longitudinal measurements were taken for 20 myelopathy patients during disease progression, and for five patients that converted to cerebral ALD during the study. Like van Ballegoij et al., the researchers found that NfL levels were significantly higher in myelopathy patients compared to controls, however in contrast to the previous study, they found no significant difference in NfL levels between asymptomatic patients and controls. As this study also included cerebral ALD patients, it was shown that NfL levels in cerebral ALD patients were significantly elevated compared to myelopathy and controls [78]. A strength of this study was the collection of longitudinal data, since the researchers found that NfL had significant prognostic ability in being able to discriminate non-converting myelopathy cases versus myelopathy patients that later converted to cerebral ALD. Additionally, in the cerebral ALD cases, a correlation was found between higher NfL levels and a higher MRI severity score (R2 = 0.73, *p* = 0.002), suggesting that NfL could reflect brain lesion severity. In support of this notion, repeated measurement of NfL in two cerebral ALD patients before and after receiving HSCT showed that NfL concentrations gradually decreased following HSCT, reflecting the arrest of neuronal damage. Lastly, the study also discovered a correlation between worsening of myelopathy as measured using the EDSS, and an increase in NfL levels, with one additional EDSS score point leading to an average 6% increase in NfL after age adjustment (95% CI: 0.7–11.4%, *p* = 0.026). This is in contrast to the results of van Ballegoij et al., who did not find any such correlation, but this disparity could be due to the very long follow-up time in the myelopathy patients of Weinhofer et al. (measurements were performed in samples collected over a period of up to 14 years), and thus the ability to observe more progression in the severity of the myelopathy [78]. From these studies, it is clear that NfL holds promise as a disease-activity biomarker for ALD which could be used to acutely monitor the response to disease-modifying treatment, however, other neurological disorders could bias measurements and therefore this would need to be incorporated into the exclusion criteria for ALD clinical trials.

## 5. Conclusions

Despite the rare occurrence of this disease, many innovative and original studies have been performed to improve knowledge surrounding molecular biomarkers for ALD. These studies generate a list of many interesting biomarker candidates (Table 1), which deserve further investigation and validation in external ALD cohorts. However, in the field there is a distinct lack of studies with longitudinal data, in addition to very small patient cohorts. This is likely due to a variety of reasons, such as the slow progression of myelopathy in ALD and thus difficulty in obtaining repeat samples over a timespan long enough to detect significant changes in biomarker levels i.e., many years.

More recently, biomarker studies often employ ultra-sensitive technology, such as single molecule array (SiMoA^TM^), to detect extremely low concentrations of the analyte of interest in biofluids. However, the integration of these systems into clinical laboratories and practice is slow, resulting in a disparity between the detection procedures used for many of these biomarker studies. For example, assays developed in-house often lack published optimization and validation data compared to those which are commercially available, resulting in many potential disease biomarkers being at different stages of development. When we look at other neurodegenerative diseases, such as Alzheimer’s disease and multiple sclerosis, in which biomarkers for early disease detection is a current hot topic and heavily researched, we can take learnings from these studies as to how certain biomarkers might be applicable to ALD. Chitinase 3-like 1, also known as YKL-40, is a microglial and/or astrocytic marker believed to reflect the neuroinflammatory process of Alzheimer’s disease. Craig-Schapiro et al. were the first to discover in two large, independent cohorts and using follow-up patient data, that the ratio of CSF YKL-40/Amyloid beta 42 measured at baseline displayed significant prognostic utility in predicting the conversion of cognitively unimpaired subjects to develop mild cognitive impairment [79]. Therefore, glial activity biomarkers, such as YKL-40 may be applicable as disease-activity biomarkers to quantify neuroinflammation in ALD and would be interesting candidates to investigate.

Overall, based off the available literature, future ALD research should investigate if an increase in MCP-1 or chitotriosidase occurs before, or as a consequence of, the neuroinflammation and damage to the BBB during the onset of cerebral ALD, and if they could be an early predictive biomarker for cerebral ALD. In addition, as shown in Alzheimer’s disease clinical trials targeting neuroinflammation [80], chitotriosidase and perhaps MCP-1 could serve as pharmacodynamic markers of neuroinflammation, specifically microglial activation, which would be relevant for use in cerebral ALD clinical trials. NfL may be suitable as a general dynamic marker of neuro-axonal injury for ALD, with longitudinal measurements being used to monitor disease activity and response to treatment in ALD clinical trials. For example, NfL levels in the blood will be affected by peripheral neuropathy, in addition to factors such as an individual’s body mass index or blood volume [81].

Thus, if found useful for ALD, future investigations will need to establish the effect of these confounding factors, including the use of age- and sex-specific reference values.

ALD is one of the most common leukodystrophies and the most common peroxisomal disease. Biomarkers that have relevance for ALD may be applicable to some of these disorders. However, there are also limitations. For example, there are currently no studies on the applicability of biomarkers in Zellweger spectrum disorders. The main difficulty is that the patient population is highly heterogeneous, both genetically and clinically. Therefore, it is not likely that biomarkers that have relevance for ALD can be applied in this group in the near future. Moreover, the lack of treatment options for Zellweger spectrum disorders makes this information less pertinent currently. However, the non-disease specific biomarkers of neuro-axonal damage, such as NfL, are more widely applicable. For instance, recently the correlation between NfL levels and MRI abnormalities in metachromatic leukodystrophy (MLD) was established [82].

As with most neurodegenerative diseases, it is likely that a panel of core biomarkers will improve the prognostic and diagnostic accuracy of ALD. Algorithms which combine different biomarkers and perhaps neuroimaging measures and other clinical biomarkers will result in greater specificity and sensitivity for ALD applications when compared to using one biomarker alone. This much-needed extensive investigation using ALD biobank samples and modelling will improve the shortcomings of the existing knowledge surrounding molecular biomarkers in ALD, in order to better monitor disease progression and have quantitative outcome measures in clinical trials.

## Figures and Tables

**Figure 1 cells-10-03427-f001:**
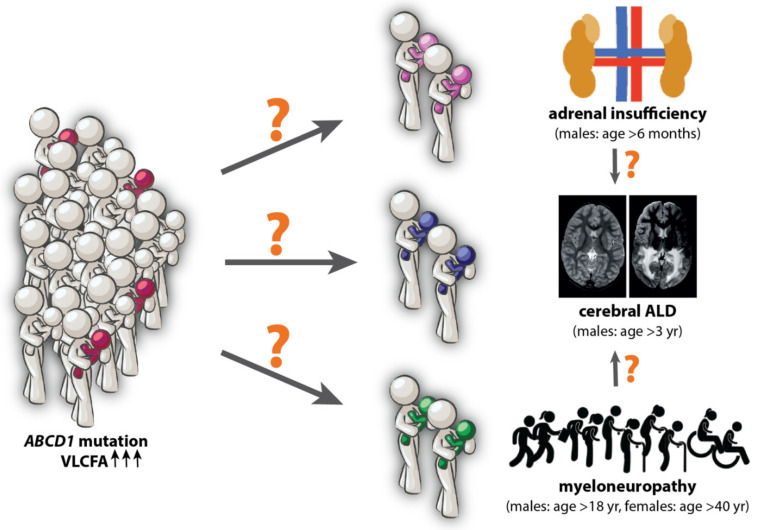
The ALD clinical spectrum. At the molecular level, ALD patients share a genetic defect in the *ABCD1* gene and elevated levels of VLCFA. ALD can be diagnosed at birth, but the clinical course cannot be predicted. Patients are pre-symptomatic at birth. In males, adrenal insufficiency is often the first symptom to appear which can be as early as six months of age. Cerebral ALD in male patients can occur at any age after three years of age. In adulthood, males and females typically develop a slowly progressive myelopathy. Male patients with myelopathy are at risk of additionally developing cerebral ALD.

**Table 1 cells-10-03427-t001:** Biomarkers examined in ALD, which could be applicable for further investigation due to the initial findings presented in the research.

Biomarker	Fluid	Change in ALD	Interpretation	Potential Application
C26:0-lysoPC	Plasma and dried blood spots	Increased in all ALD phenotypes, including ALD women with normal plasma VLCFA levels.	Data is required to investigate correlations with disease severity.	As a diagnostic biomarker in newborns, male and female ALD patients.
Reduced Glutathione (GSH)	Plasma	Decreased in all ALD phenotypes compared to controls.	Data is required to investigate correlations with disease severity and longitudinal changes in GSH levels.	As a disease-activity biomarker to quantify oxidative stress.
Monocyte Chemoattractant Protein 1 (MCP1)	CSF	Increased in cerebral ALD patients compared to controls.	MCP1 correlates with MRI severity score, chitotriosidase activity and total protein levels. Longitudinal data over the course of treatment is needed.	As a screening biomarker to detect neuroinflammation. As a prognostic biomarker to predict post-treatment outcome.
Total protein levels	CSF	Increased in cerebral ALD patients compared to controls.	Total protein levels strongly correlate to pre-transplant MRI severity scores and pre- and post- HSCT neurological functional scores.	As a prognostic biomarker to predict post-treatment outcome.
Chitotriosidase	CSF and plasma	Increased in cerebral ALD patients compared to controls.	Chitotriosidase correlates with pre- and post- HSCT neurological functional scores, post-transplant MRI severity score and the change in functional status.	As a screening biomarker to detect neuroinflammation. As a prognostic biomarker to predict post-treatment outcome.
Superoxide dismutase (SOD)	Plasma	SOD activity decreases in a stepwise manner for ALD phenotypes; control > myelopathy > cerebral ALD	SOD activity inversely correlates with clinical MRI severity score in cerebral ALD patients. Longitudinal data to the onset of cerebral ALD is needed.	As a disease-activity biomarker to quantify oxidative stress. As a screening biomarker to detect the onset of cerebral ALD.
Autoantibodies against profilin 1 (PFN1)	Plasma	Increased in boys with cerebral ALD compared to non-cerebral cases and controls.	Longitudinal data pre- and post-onset of cerebral ALD and over the course of HSCT treatment is needed.	As a screening biomarker to detect the onset of cerebral ALD.
Neurofilament light chain (NfL)	Plasma	Increased in ALD patients compared to controls. Increased in boys with cerebral ALD compared to non-cerebral cases and controls.	NfL correlates with clinical measures of myelopathy severity in male ALD patients, and with MRI severity score in cerebral ALD patients.	As a prognostic biomarker to predict the onset of cerebral ALD. As a treatment-responsive biomarker to reflect neuronal damage.

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
