# Peer review of "Molecular Biomarkers for Adrenoleukodystrophy: An Unmet Need"

_cells, 2021, doi:10.3390/cells10123427_

Round 1

Reviewer 1 Report

The review provides a comprehensive discussion of the state of the field regarding the identification of biomarkers for the peroxisomal disease adrenoleukodystrophy.  The authors correctly make the point that the identification of biomarkers are essential for early disease diagnosis but also for predicting clinical prognoses leading to improved and more target therapy. In addition to summarizing the current work regarding biomarkers, the authors also pinpoint noticeable deficiencies including the paucity of longitudinal studies and the lack of available large cohort for study. 

Although this is a well written and comprehensive review, I have 2 minor suggestions:

  1. The authors mention that ALD is the consequence of mutations in the ABCD1 gene.  It would be informative to the reader to have more details of the known mutations.  What type of of mutations are involved (eg frame shift, missense and nonsense?) and is there any correlation between disease severity of the type of mutation?  Also, can a patient have more than one mutation in the ABCD1 gene and if so, is there correlation with the number of mutations and disease severity?
  2. Although the authors make the point that there is no correlation between disease severity and VLCFA accumulation, a recent paper presents data that suggests that such a correlation may exist.  The authors should consider including this findings from this recent paper (Fujiwara et al., 2021 Int J Mol Sci 22(16):8645).  In this paper the authors provide a potential correlation between the levels of certain species of C26 VLCFAs and forms of ALD.

Author Response

We thank the reviewer for the kind words and suggestions, which we address below

Reviewer: The authors mention that ALD is the consequence of mutations in the ABCD1 gene. It would be informative to the reader to have more details of the known mutations. What type of mutations are involved (eg frame shift, missense and nonsense?) and is there any correlation between disease severity of the type of mutation?

Answer: We have another manuscript under review for the same special edition on “Peroxisomal Disorders: Development of Targeted Therapies”. The title of that manuscript is: “Structure and function of the adrenoleukodystrophy mutation database: 20 years, 920 mutations and 3000 cases”. That manuscript is completely focused on the question raised by the reviewer.

We extended the section on genotype-phenotype to address the reviewers’ question. Lines 74-80: “All patients have a mutation in the ABCD1 gene with mutations ranging from missense mutations to large deletions. Despite being a monogenetic disease, mutations in the ABCD1 gene have no predictive value with respect to clinical outcome. The lack of a simple genotype-phenotype correlation can be exemplified with a case study detailing six brothers with an identical p.Pro484Arg ABCD1 missense mutation, yet presenting with five different ALD phenotypes (ranging from cerebral ALD in childhood to adrenal insufficiency in adulthood) in the family [14].”

Reviewer: Also, can a patient have more than one mutation in the ABCD1 gene and if so, is there correlation with the number of mutations and disease severity?

Answer: Several patients have indeed been reported with two missense variants in a single allele (for example p.[Pro218Ser; Val222Leu]), but also with two nonsense mutations in a single allele (p.[Gln430*; Arg464*]). These patients are not more severely affected than patients with a single missense mutation or exome deletion, which is in line with the lack of a genotype-phenotype correlation.

Reviewer: Although the authors make the point that there is no correlation between disease severity and VLCFA accumulation, a recent paper presents data that suggests that such a correlation may exist. The authors should consider including this findings from this recent paper (Fujiwara et al., 2021 Int J Mol Sci 22(16):8645). In this paper the authors provide a potential correlation between the levels of certain species of C26 VLCFAs and forms of ALD.

Answer: We have added the Fujiwara study to the complex lipids (4.2) section (lines 232-239). The preliminary results of this study warrant further investigation. But it is too soon to conclude that there may be a correlation between disease severity and specific lipids, because the study lacks details clinical information (especially the age of AMN patients as AMN patients are still at risk of developing cerebral ALD) and the sample size of 3 and 4 per phenotype is very small. We thank the reviewer for pointing out this study.

Reviewer 2 Report

The review by Honey et al. presents in a very clear and comprehensive manner the current knowledge on biomarkers related to X-linked adrenoleukodystrophy. The list of references seems to me complete and appropriate.

I would recommend that the authors add a small paragraph about existing biobanks and/or the need to create specific biobanks that would facilitate posterior studies of current markers to consolidate data or researches for potential new biomarkers.

As the authors point out, long-term storage may be a serious concern, especially when oxidative markers are searched but in many cases, the availability of samples should be useful. Inded, the need for longitudinal studies and the need to increase sample size (here again, I agree with the comments of the authors on these points) are huge when initiating a research project aiming at identifying X-ALD biomarkers. Recruiting of patients in a rare disease is somehow difficult.

Minor comments:

  • I would recommend removing the reference to unpublished data in lane 196. There is indeed no evidence that C26:0-lysoPC can be used for patient monitoring.
  • Please improve the formatting of the table. The text in the different cells can probaly be shortened a bit.

Author Response

Answer: we thank the reviewer for the kind words and suggestions, which we address below

Reviewer: I would recommend that the authors add a small paragraph about existing biobanks and/or the need to create specific biobanks that would facilitate posterior studies of current markers to consolidate data or researches for potential new biomarkers.

Answer: We thank the reviewer for this suggestion. We have extended the section “Setting the stage” (lines 129-138) with “Biobanks only have value for this kind of research if samples are collected and stored in a standardized way in a prospective manner and can be linked to detailed and accurate clinical information of individual patients. Although for ALD a few of these cohorts exist, these are all local efforts in individual centers. Larger international cohorts and biobanks would be preferable, but have not materialized even though researchers definitely recognize the potential benefits. Major obstacles are complex regulatory issues and ownership of samples. An alternative would be to maintain individual cohorts and biobanks, but with a standardized follow-up protocol to make the individual data sets more compatible. It will be interesting to see how the field will work towards better collaboration.”

Reviewer: As the authors point out, long-term storage may be a serious concern, especially when oxidative markers are searched but in many cases, the availability of samples should be useful. Inded, the need for longitudinal studies and the need to increase sample size (here again, I agree with the comments of the authors on these points) are huge when initiating a research project aiming at identifying X-ALD biomarkers. Recruiting of patients in a rare disease is somehow difficult.

Answer: We thank you reviewer for acknowledging these concerns. Storage conditions, collaboration, external validation are typical topics that we discuss with our (international) colleagues when talking about biomarker research and sharing samples. We did address these concerns in the previous point.

Minor comments:

Reviewer: I would recommend removing the reference to unpublished data in lane 196. There is indeed no evidence that C26:0-lysoPC can be used for patient monitoring.

Answer: (lines 206-209) we removed the reference to unpublished data and changed the sentence to “Currently there is no evidence that C26:0-lysoPC levels in plasma or dried blood spots is predictive concerning disease progression or manifestations. Whether C26:0-lysoPC levels in brain correlate with phenotype, as has been demonstrated for VLCFA levels [32], has not yet been investigated.”

Reviewer: Please improve the formatting of the table. The text in the different cells can probaly be shortened a bit.

Answer: We agree with the reviewer that the Table could be shortened a bit. We shortened the text in columns “Change in ALD” and “Interpretation”. And in response to the comments made by reviewer 3 we aligned the wording in the column “potential application” with the wording used in section 2 “…new ALD biomarkers are needed:” followed by 4 points and reasons: “As a diagnostic marker, As a predictive marker, As a prognostic biomarker, As a disease monitoring biomarker”. 

Reviewer 3 Report

The manuscript by by Honey et al. reviews the current status of biomarkers in X-linked adrenoleukodystrophy. Several categories of biomarker, such as diagnostic, prognostic, therapeutic, etc. are evaluated. Their pros and cons are well-described, and their robustness and potential clinical uses are discussed. Needs for future investigations for each biomarker are thoughtfully laid out. A table nicely summarizes the findings and succinctly presents the reader with an easy to understand picture of the field. 

This reviewer has no major concerns or criticisms.

Minor issues:

  1. Line 76 – is the correct term “monogenic” vs. “monogenetic”? Please check and change if necessary
  2. In lines 86-87, the authors write “…new ALD biomarkers are needed:” followed by 4 points. The first two points (lines 88 and 93) begin with “As a…”  For grammatical consistency, the last 2 points (lines 99 and 102) should also begin with “As a…”
  3. Line 277 – the abbreviations CML, CEL, MDAL, and AASA should be defined
  4. Line 493 – Shouldn’t “increased cerebral” be “increased in cerebral”?

Author Response

We thank the reviewer for the very kind words and pointing out the minor issues, which we address below.

Reviewer: Line 76 – is the correct term “monogenic” vs. “monogenetic”? Please check and change if necessary

Answer: the term monogenic refers to the single gene involved in the expression of a trait/disease. As there is a single gene responsible for ALD, monogenic seems correct. However, we did want to double/triple fact-check this :-)

A single-gene disorder is a monogenic disorder.

Monogenetic is related to monogenesis and is a term that is used in evolution-related genetics/topics: “origin of diverse individuals or kinds (as of language) by descent from a single ancestral individual or kind” (Merriam Webster).

Reviewer: In lines 86-87, the authors write “…new ALD biomarkers are needed:” followed by 4 points. The first two points (lines 88 and 93) begin with “As a…”  For grammatical consistency, the last 2 points (lines 99 and 102) should also begin with “As a…”

Answer: thank you reviewer! We really appreciate this. We changed points 3 and 4 to “As a prognostic biomarker to predict ….” And “As a disease monitoring biomarker that correlates ….” (Lines 100 and 103).

In response to a question raised by reviewer 2, we aligned the wording used in this section with the wording used in the Table, column “potential application”, which now also uses: “As a diagnostic marker, As a predictive marker, As a prognostic biomarker, As a disease monitoring biomarker”.

Reviewer: Line 277 – the abbreviations CML, CEL, MDAL, and AASA should be defined

Answer: We defined the abbreviations “(Ne-carboxymethyl-lysine (CML), Ne-carboxyethyl-lysine (CEL), Ne-malondialdehyde-lysine (MDAL) and aminoadipic semialdehyde (AASA)” (Lines 296-297)

Reviewer: Line 493 – Shouldn’t “increased cerebral” be “increased in cerebral”?

Answer: Thank you reviewer. Indeed, here we missed an “in” (line 511)